# Preparation and Application of Fe-N Co-Doped GNR@CNT Cathode Oxygen Reduction Reaction Catalyst in Microbial Fuel Cells

**DOI:** 10.3390/nano11020377

**Published:** 2021-02-02

**Authors:** Man Zhang, Zhaokun Ma, Huaihe Song

**Affiliations:** Beijing Key Laboratory of Electrochemical Process and Technology for Materials, Beijing University of Chemical Technology, Beijing 100029, China; 2018400229@mail.buct.edu.cn

**Keywords:** microbial fuel cells, oxygen reduction reaction, graphene nanoribbon@carbon nanotube, graphene nanoribbon

## Abstract

Through one-step pyrolysis, non-noble-metal oxygen reduction reaction (ORR) electrocatalysts were constructed from ferric trichloride, melamine, and graphene nanoribbon@carbon nanotube (GNR@CNT), in which a portion of the multiwall carbon nanotube is unwrapped/unzipped radially, and thus graphene nanoribbon is exposed. In this study, Fe-N/GNR@CNT materials were used as an air-cathode electrocatalyst in microbial fuel cells (MFCs) for the first time. The Fe-N/C shows similar power generation ability to commercial Pt/C, and its electron transfer number is 3.57, indicating that the ORR process primarily occurs with 4-electron. Fe species, pyridinic-N, graphitic-N, and oxygen-containing groups existing in GNR@CNT frameworks are likely to endow the electrocatalysts with good ORR performance, suggesting that a GNR@CNT-based carbon supporter would be a good candidate for the non-precious metal catalyst to replace Pt-based precious metal.

## 1. Introduction

Microbial fuel cells (MFCs) enable the degradation of sewage and produce useful electricity at the same time [1,2]. Therefore, they have created a research upsurge in the field of energy storage in recent years. Since oxygen is widely available and the product of the oxygen reduction reaction (ORR) is generally water, which does not cause secondary pollution, air-cathode MFC is considered a promising configuration. Until now, the low energy output has been the bottleneck restricting the practical application of MFCs [3].

Several factors affect MFC performance, including anode materials [4,5,6], MFC configurations [7,8,9], cathode catalysts [10,11,12], and electroactive bacteria [13,14]. Among them, the high overpotential loss and price of ORR catalysts play an important role in limiting the power generation capacity. Platinum (Pt)-based materials are still the most efficient ORR catalysts at present, despite their shortcomings, such as low reserves, high cost, poor durability, and inactivation caused by poisoning. Given these disadvantages of the noble metal catalysts, numerous scientists are exploring advanced, low-cost materials to replace Pt-based materials [15,16,17]. In particular, carbon-based electrocatalysts containing Fe and N elements (Fe-N/C) have been widely used as the most promising alternatives to noble metal Pt, due to the similar catalytic activity and abundant precursor species [18,19]. Pyridinic-N is commonly recognized as the dominant active site to catalyze the oxygen reduction reaction, and Fe-N_x_ species have also been suggested as active sites [20,21,22]. The catalytic mechanism and the precise active sites are still not fundamentally understood, owing to the heterogeneous moieties and surrounding environments within the electrocatalysts [22].

Generally speaking, Fe-N/C-based catalysts can be obtained by the direct pyrolysis of the precursors containing Fe, N, as well as C, at 600–1000 °C. In this way, a large amount of Fe and N co-doped materials equipped with considerable ORR catalytic properties were prepared [23,24,25,26]. Considering the unique physical and chemical properties of different carbon sources, such as morphology, conductivity, and specific surface area, carbon supporters have an important impact on the ORR at the triple-phase interface [27]. A range of carbon materials have been explored, containing covalent triazine framework [28], hierarchically porous carbon [29], carbon nanosheet [30], carbon nanotube (CNT) [31], graphene [32], etc. For example, Wang et al. prepared Fe-N co-doped hierarchical porous 2D carbon nanosheets using a top-down strategy. They showed high ORR activity when used as a cathode catalyst in an alkaline medium [30]. Kang and coworkers constructed Fe-N/CNT material via a direct growth method [33]. The prepared catalyst had a one-dimensional (1D) framework to increase electron transfer, and a large specific surface area to load the active sites, leading to good oxygen reduction ability in acid solution. Moreover, graphene is also suitable for catalytic applications because of its good conductivity and rich edge defects. Ma and his research group chose three kinds of graphene, including high-activity graphene (HAG), high-conductivity graphene (HCG), and single-layer graphene (SLG), as the carbon source to compare their performance [34]. They found that multiwall carbon nanotube (MW-CNT) were grown during the preparation of Fe-based catalysts using single-layer graphene as a carbon source. Both the onset potential (E_onset_) and the half-wave potential (E_1/2_) of the obtained Fe-N/SLG nanocomposite were superior to that of Pt/C.

Graphene nanoribbon (GNR) is thin elongated strips of sp^2^-hybridized carbon atoms with quasi-one-dimensional structures [35]. Correspondingly, GNRs have a large aspect ratio/length–width ratio, and an abundance of non-tri-coordinate carbon atoms at the edges as well as defects, which is favorable for improving the density of reactive sites [36]. The variety of the edges of GNR results in different chemical and electronic characteristics, such as zigzag-edge and armchair-edge. Generally speaking, the easiest way to produce GNR is by longitudinally unwrapping carbon nanotube [37,38,39]. The edges can act as sites to anchor heteroatoms. It was reported that the doping amount of heteroatoms is positively correlated with the edge level [40,41]. Nevertheless, the conductivity of the material will inevitably be damaged. The pure GNR can more feasibly be restacked, resulting in lower utilization of reactive sites. Thus, the oxygen reduction activity of the electrocatalysts prepared from fully opened MW-CNT (pristine graphene nanoribbon) is not always satisfactory, as demonstrated by Shui et al. [42]. Accordingly, several heteroatom-doped partially unwrapped MW-CNT along the radical axis (GNR@CNT) have been explored as ORR catalysts, such as self-doping (O), N doping, [36], N, S co-doping, [40], and Fe, N co-doping [43]. Shui and his coworkers discovered that the zigzag carbon atom is the most active catalytic place among all of the carbon defects on GNR@CNT for the ORR in an alkaline medium for proton exchange membrane fuel cells. Zhou et al. prepared an Fe-N co-doped GNR@CNT composite to illustrate that longitudinally unwrapping MW-CNT material is rich in edge defects which can augment both the Fe-N-doping degree and reactive sites during the pyrolysis process [43]. The obtained FeN-uCNT-700 showed good oxygen reduction properties and durability in alkaline solution. Some of these catalysts revealed good performance in acidic or alkaline medium.

However, graphene-nanoribbon-based materials have not yet been studied in neutral media for microbial fuel cells. Within the scope of laboratory research, the nutrient of the MFCs usually consists of vitamins, minerals, phosphate buffer, and organic matter. In terms of practical application, the composition of municipal sewage is more complicated. In addition, electrogenic microorganisms not only produce some metabolites, but also adhere to the cathode surface to form biofilms, which affect the catalytic activity of the electrocatalysts. Thus, the electrolyte compositions of the MFCs are more complex than the acid or alkaline solutions used in other types of fuel cells.

Considering the above, GNR@CNT was first used as a carbon support herein to prepare cathode catalysts for MFCs. Accordingly, we fabricated Fe-N co-doped GNR@CNT (denoted as Fe-N/GNR@CNT) via one-pot pyrolysis, which served as a cathode catalyst to study its oxygen reduction reaction performance in a neutral solution for MFCs.

## 2. Materials and Methods

### 2.1. Synthesis of Fe-N/GNR@CNT Electrocatalysts

In this study, the GNR@CNT precursor was prepared by partially unwrapping multiwall carbon nanotubes in the longitudinal direction, as shown in Figure 1. The concrete step was the same as reported in the literature [42]. Firstly, we measured out 400 mg of pristine MW-CNTs (0.5–2 μm in length and 30–50 nm in outer diameter) into a beaker, and then added 72 mL of concentrated H_2_SO_4_ into the reaction system. Secondly, after stirring evenly, we slowly added 1.6 g KMnO_4_ into the above reaction vessel and continued stirring for 1 h. The process of adding KMnO_4_ required an ice bath. It was placed in a 55 °C water bath for 15 min, then transferred to a 70 °C water bath for 45 min. We stopped heating, and cooled down the mixture to room temperature. Subsequently, it was poured into 200 mL of iced water with 6 mL of hydrogen peroxide. It was centrifuged at 10,000 rpm, and the impurities were removed with diluted hydrochloric acid. Finally, after several days of dialysis, the oxidized GNR@CNT precursor was obtained.

The Fe-N/C materials with different Fe-doping ratios were prepared as follows: 100 mg of oxidized GNR@CNT and 100 mg of melamine were mixed with different amounts of FeCl_3_ (containing 25, 50, and 100 mg, respectively), followed by carbonization at 800 °C for 60 min in Ar atmosphere. Accordingly, they were denoted as Fe-N/C-1:4:4, Fe-N/C-1:2:2, and Fe-N/C-1:1:1, respectively. The MFCs using Fe-N/C-1:4:4, Fe-N/C-1:2:2, and Fe-N/C-1:1:1 materials as cathode catalysts were called MFC-Fe-N/C-1:4:4, MFC-Fe-N/C-1:2:2, and MFC-Fe-N/C-1:1:1, respectively.

Fe-N/C materials with different N-doping ratios were also synthesized. Oxidized GNR@CNT (100 mg) and 25 mg of FeCl_3_ were mixed with different amounts of melamine (containing 50, 200, and 400 mg, respectively), and the other steps were all the same as above. Accordingly, they were denoted as Fe-N/C-1:2:4, Fe-N/C-1:8:4, and Fe-N/C-1:16:4, respectively. The MFCs using Fe-N/C-1:2:4, Fe-N/C-1:8:4, and Fe-N/C-1:16:4 materials as cathode catalysts were named as MFC-Fe-N/C-1:2:4, MFC-Fe-N/C-1:8:4, and MFC-Fe-N/C-1:16:4, respectively.

### 2.2. MFC Construction and Setup

A single-chamber air-cathode MFC (Physics&Chemistry Co., Ltd., Kowloon, HK) was adopted as a reaction device, and the effective area of the carbon cloth cathode was 7 cm^2^. Detailed parameters can be found in the literature [44,45]. The catalyst loadings were all the same (2 mg·cm^−2^). Acid-treated carbon fiber brushes were used as an anode to attach microorganisms. The pitch-based carbon fibers were cut to 2.5 cm long, then weighed out to 500 mg of carbon fibers. The fibers were fixed and twisted with titanium wires.

Additionally, the external resistance was fixed at 1 kΩ. The sludge with mixed cultures came from the Gaobeidian Wastewater Treatment Plant, Beijing. For the battery setup phase, a 1:1 volume ratio of sewage and nutrients was added, and the medium was changed every 24 h. Each MFC was tested in parallel and kept at about 30 °C.

### 2.3. Analysis and Measures

The output voltages of each MFC loading with a 1 kΩ resister were measured by a data acquisition system (CT2001A, LANHE). The polarization data of the MFCs were tested by the pseudo-steady-state discharge method. The external resistors were decreased from 10 kΩ to 100 Ω, one step at a time. For each resistance, the potentials between anode and cathode, anode and reference electrode (Ag/AgCl), and cathode and reference electrode were measured every 10 min under transient-steady-state circumstances by a multitester. Furthermore, the slope of the linear part of the polarization curves was equivalent to the apparent internal resistance (R_int_) of the MFCs.

The crystal structures of the materials were obtained by X-ray diffraction (XRD) spectra. The structure and morphology of the GNR@CNT-based composite were tested by scanning electron microscopy (SEM) and transmission electron microscopy (TEM). The graphitization degree of the catalyst was characterized by Raman spectra analysis. The functional groups on the surface of the sample were tested using X-ray photoelectron spectroscopy (XPS). The specific surface area was determined using the Brunauer-Emmett-Teller(BET) adsorption isotherm method.

The ORR catalytic activities were characterized by a rotating disk electrode (RDE) test. The reference electrode was Ag/AgCl, and the counter electrode was a carbon rod. For the working electrode, 5 mg of the catalysts was dispersed in 1 mL ethanol by the ultrasonic meter. Then, 10 μL of 5% Nafion was added and continued with sonication for 60 min to prepare a homogeneous suspension. The medium was 50 mM PBS (pH = 7).Catalyst ink (10 μL) was dripped to the working electrode. After drying, ORR tests were performed. The electron transfer number (*n*) towards the ORR was calculated from the Koutecky-Levich(K–L) plot:1/J = 1/J_K_ + 1/J_L_ = 1/J_K_ + 1/(Bω^1/2^)(1)
B = 0.62nFAC_0_(D_0_)^2/3^ν^−1/6^(2)
where J, J_K_, and J_L_ represent the measured, kinetic, and diffusion-limited current density, respectively; ω is the rotating speed of the working elctrode; F is the Faraday constant; A is the area of the working electrode; D_0_ is the diffusion co-efficiency of oxygen; C_0_ is the bulk concentration of the dissolved oxygen in the solution; ν is the kinematic viscosity of the solution.

## 3. Results

### 3.1. The Identification of GNR@CNT

In Figure 2A–C, we can observe that the bare, multiwalled CNT materials were partially unzipped to form GNRs after being treated with concentrated H_2_SO_4_ and KMnO_4_. The GNR@CNT sample features rich edge contents. Compared with the pristine CNT, as can be seen from Figure 2D, the graphitic (002) peaks at 26° and (100) at 43° of the GNR@CNT hybrid become weakened, and the graphene-oxide (001) peak at about 10° appears. Meanwhile, as can be seen from Appendix A, the oxygen content of GNR@CNT is increased from 2.54% to 38.28%. Oxygen-containing groups may help anchor iron ions and melamine, which is beneficial to improve the heteroatom-doping level of the catalysts. The increased sulfur content comes from the concentrated H_2_SO_4_.

Moreover, the destruction of the outside wall of the original CNT can also be demonstrated by Raman spectra (Figure 2E). The I_D_/I_G_ ratio of pure CNT was 1.02. The unzipped sample shows an increased I_D_/I_G_ ratio (1.14), indicating the reduced graphitization degree as well as the successful exfoliation of the CNT.

### 3.2. The Performance of the Fe-N/C Materials with Different Fe-Doping Contents

To explore the appropriate Fe-doping contents, we fixed the ratio of carbon support to melamine at 1:1 and gradually increased the additive content of FeCl_3_. XRD patterns (Figure 3) indicate that the oxidized GNR@CNT substrates were reduced to GNR@CNT with the graphene-oxide (001) peak vanishing. At the same time, the characteristic peaks of Fe_3_C nanoparticles emerged after heat treatment. Meanwhile, with the increase of FeCl_3_, more and more Fe_3_C nanoparticles were formed (Appendix A). According to the previous research results, metallic Fe/Fe_3_C provided good ORR activity, and the Fe-based nanoparticles were able to increase the catalytic ability of the N-C_x_ active sites due to the reduction of the working function that arises from the electrons transfer from Fe or Fe_3_C nanoparticles to the carbon planes [46,47,48].

The maximum power densities of MFC-Fe-N/C-1:4:4, MFC-Fe-N/C-1:2:2, and MFC-Fe-N/C-1:1:1 were about 801, 701, and 454 mW/m^2^, respectively (Figure 4A). With the ratio of FeCl_3_ increasing, the MFC performance worsened. From the slope of the polarization curves (Figure 4B), the internal resistances of MFC-Fe-N/C-1:4:4, MFC-Fe-N/C-1:2:2, and MFC-Fe-N/C-1:1:1 were ca. 193, 207, and 249 Ω, respectively. The potentials of the anodes are similar, so the difference in MFC performance was mainly brought about by the cathode catalysts (Figure 4C).

### 3.3. The Performance of the Fe-N/C Materials with Different N-Doping Contents

#### 3.3.1. The Performance of Fe-N/C Catalysts with Different N-Doping Amounts

Subsequently, we prepared electrocatalysts with different N-doping amounts to explore the bioelectricity generation performance. In Appendix A, it can be seen that in addition to the characteristic peaks of amorphous carbon, the characteristic peaks of Fe_3_C were also present in the Fe-based catalysts with different nitrogen-doping amounts. After carbonization, aggregates of carbon nanotube derivatives were visible, and Fe-based nanoparticles could also be observed growing on the surface of the carbon matrix (Appendix A). Meanwhile, the EDS mapping images of Fe-N/C-1:4:4 show that O, N, Fe, and S elements were evenly distributed on the surface (Appendix A), indicating the successful doping of heteroatoms.

All the MFCs were set up successfully and the output voltages under 1000 Ω external resistance of MFC-Pt/C, MFC-Fe-N/C-1:2:4, MFC-Fe-N/C-1:4:4, MFC-Fe-N/C-1:8:4, and MFC-Fe-N/C-1:16:4 were approximately 547, 460, 580, 560, and 460 mV, respectively (Figure 5A). The maximum power densities of MFC-Pt/C, MFC-Fe-N/C-1:2:4, MFC-Fe-N/C-1:4:4, MFC-Fe-N/C-1:8:4, and MFC-Fe-N/C-1:16:4 after cycling for one week were ca. 851, 590, 801, 752, and 590 mW/m^2^, respectively (Figure 5B). The Fe-N/C-1:4:4 sample possessed the best electricity generation performance, which is only 5.6% less than that of Pt/C. From the slope of the ohmic polarization region (Figure 5C), we can see that the internal resistances of these MFCs were 183, 239, 193, 203, and 229 Ω, respectively. The anode polarization curves (Figure 5D) indicate that the potentials of the anodes were similar. All conditions are the same except for the cathode catalysts, so the difference in battery performance mainly came from the cathode electrocatalysts. Furthermore, we also tested the cycling stability of Pt/C and MFC-Fe-N/C-1:4:4. After cycling for four weeks, the power densities of MFC-Pt/C and MFC-Fe-N/C-1:4:4 were reduced to 722 and 704 mW/m^2^, respectively (Appendix A), suggesting similar cyclic stability.

#### 3.3.2. XPS and BET Specific Surface Area Analysis

XPS measurements were further carried out to analyze the elemental composition and chemical states on the surface of these catalysts with different N-doping contents. XPS survey spectra of the as-obtained Fe-N/C samples also confirm the coexistence of C, N, O, Fe, and S elements on the samples’ surfaces (Appendix A). The ratio values of every kind of element are shown in Table 1. The Fe-N/C-1:4:4 sample exhibited the highest surface heteroatomic doping contents including N, O, Fe, and S. The N 1s spectra show an enlarged peak, which is deconvoluted in four peaks (i.e., 398.1, 399.3, 400.9, and 403.4 eV) related to pyridinic-N, pyrrolic-N, graphitic-N, and oxidized-N, respectively (Figure 6). The ratio values of each nitrogen species are summarized in Table 1. The Fe-N/C-1:4:4 sample not only exhibited the highest nitrogen-doping content but also showed the highest amount of pyridinic-N. The graphitic-N content in Fe-N/C-1:4:4 was also higher. According to the previous reports, pyridinic-N and graphitic-N may be the catalytic sites for the ORR [43,49]. The peak at ~285.8 eV indicates the C-N/C-S bonds in the fitted high-resolution C 1s spectra (Appendix A). The O 1s XPS spectra are deconvoluted in two peaks related to C=O/O-C=O at 531.1 eV and C-O-H/C-O-C at ~532.8 eV (Appendix A). Previous literature presumed that oxygen-containing functionalized species in the catalysts were likely to reduce the activation energy of the oxygen reduction process in neutral electrolyte [50,51]. The oxygen groups could make the catalyst surface more negative, which may serve as the active center to load iron atoms, comprising adsorption, coordination, and replacement [34]. The Fe 2p HR-XPS spectra in Appendix A demonstrate the coexistence of Fe^0^ and Fe^2+^/Fe^3+^ [52]. Ferric ions can combine with pyridinic-N to form Fe-N_x_ species, which have better ORR catalytic activity. In short, the coexistence of active N and Fe species with relatively high O contents may give the Fe-N/C-1:4:4 catalyst a good catalytic activity [53].

The specific surface area of Fe-N/C electrocatalysts with different nitrogen-doping content was measured by nitrogen adsorption–desorption isotherms. All of the Fe-based catalysts exhibited type IV isotherms with an obvious hysteresis loop at P/P_0_ ranging from 0.4 to 1.0 (Figure 7A) [34]. This phenomenon is the typical symbol of mesoporous structure formed by the stacking of particles. Furthermore, at low relative pressure, the adsorption capacities increased rapidly, indicating that there were micropores in the materials as well. The pore size distribution curves (Figure 7B) also demonstrate that the pores in the materials were mainly micropores and mesopores. Pore channels guarantee the diffusion of electrolyte and oxygen to the active sites. The specific surface areas of Fe-N/C-1:2:4, Fe-N/C-1:4:4, Fe-N/C-1:8:4, and Fe-N/C-1:16:4 were 352.3, 197.4, 73.6, and 237 m^2^/g, respectively (Table 2), indicating that the ORR electrocatalytic activity of the Fe-N/C-1:4:4 sample was dominated by other factors, such as the heteroatom doping level and number of active sites.

#### 3.3.3. RDE Tests

To study the catalytic characteristics of the Fe-N/C-1:4:4, an RDE test was conducted in 50 mM PBS electrolyte. Pt/C was measured for comparison as well. The E_1/2_ of Fe-N/C-1:4:4 was more negative than Pt/C (Figure 8A). The results of the RDE test corresponded with the MFC performance. Figure 8B,C display the linear sweep voltammetry(LSV) plots of the Fe-N/C-1:4:4 and Pt/C at different rotating speeds. From the slope of K–L plots, we can obtain the electron transfer numbers of Fe-N/C-1:4:4 and Pt/C were ca. 3.57 and 3.90, respectively, suggesting that Fe-N/C-1:4:4 mainly catalyzed the ORR process by a 4 e^−^ reduction pathway.

## 4. Discussion

In summary, we synthesized GNR@CNT materials with abundant edges by chemical oxidation. Fe-N/GNR@CNT showed good catalytic activity towards oxygen reduction in neutral electrolyte, which can be attributed to the synergistic effect of Fe_3_C nanoparticles, pyridinic- and graphitic-N groups, and oxygen-containing species existing in the matrix of GNR@CNT. The MFC using Fe-N/C-1:4:4 as the cathode catalyst exhibited good bioelectricity generation ability, and it mainly catalyzed the oxygen reduction reaction procedure by the four-electron pathway. Although the MFC performance and catalytic activity of the Fe-N/GNR@CNT sample were slightly inferior to that of Pt/C, the cost of the GNR@CNT-based materials shows great potential for real-world application.

## Figures and Tables

**Figure 1 nanomaterials-11-00377-f001:**
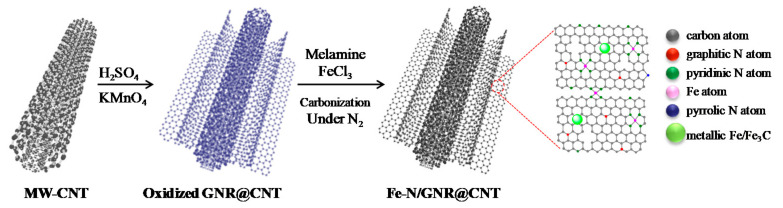
Illustration of the preparation process for Fe-N co-doped graphene nanoribbon@carbon nanotube (GNR@CNT precursor).

**Figure 2 nanomaterials-11-00377-f002:**
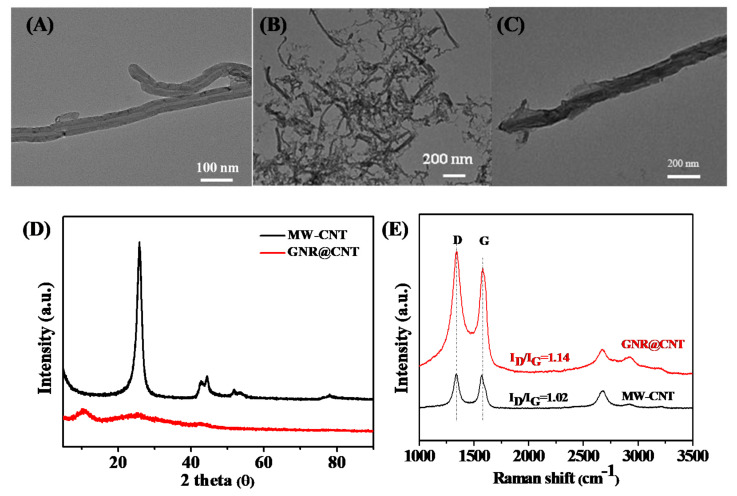
TEM images of: (**A**) pure multiwall carbon nanotube (MW-CNT) and (**B**,**C**) GNR@CNT; (**D**) XRD patterns of pure CNT and GNR@CNT; (**E**) Raman spectra of pristine CNT and GNR@CNT.

**Figure 3 nanomaterials-11-00377-f003:**
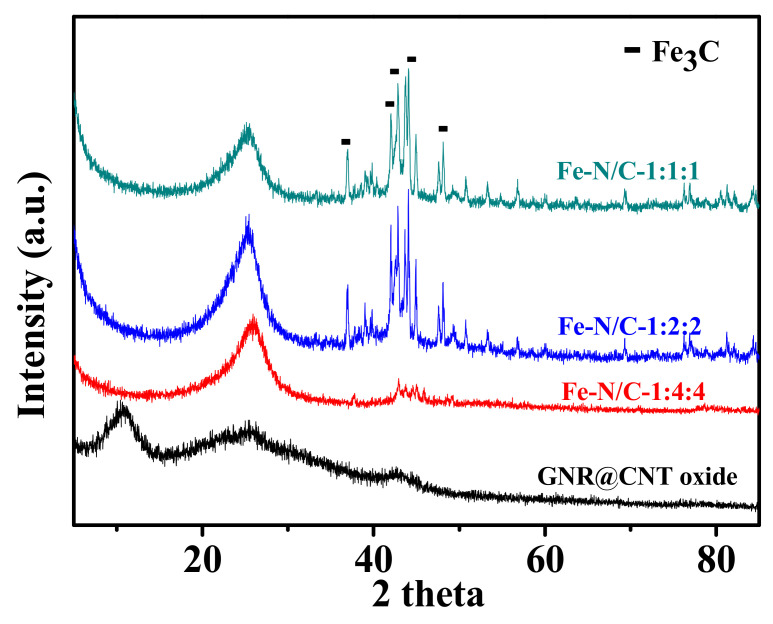
XRD patterns of Fe-N/GNR@CNT samples changing with the FeCl_3_ doping content.

**Figure 4 nanomaterials-11-00377-f004:**
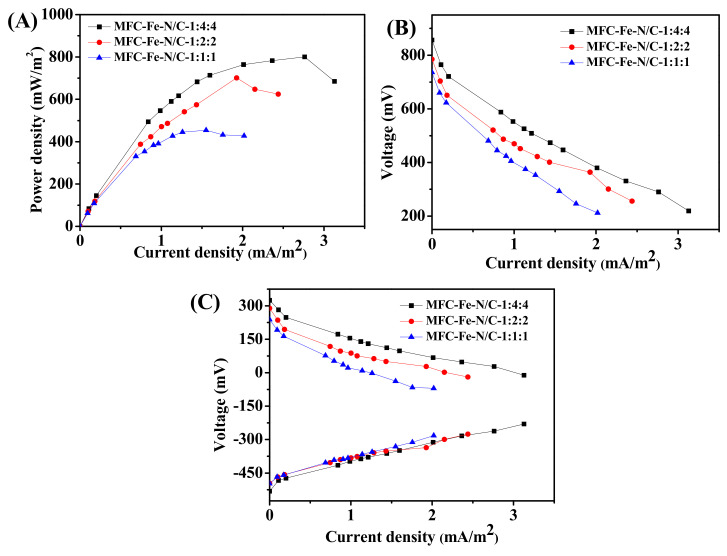
Electricity production performance of the microbial fuel cells (MFCs) using the catalysts with different Fe-doping contents: (**A**) power density curves; (**B**) polarization curves; and (**C**) the anode and cathode polarization curves.

**Figure 5 nanomaterials-11-00377-f005:**
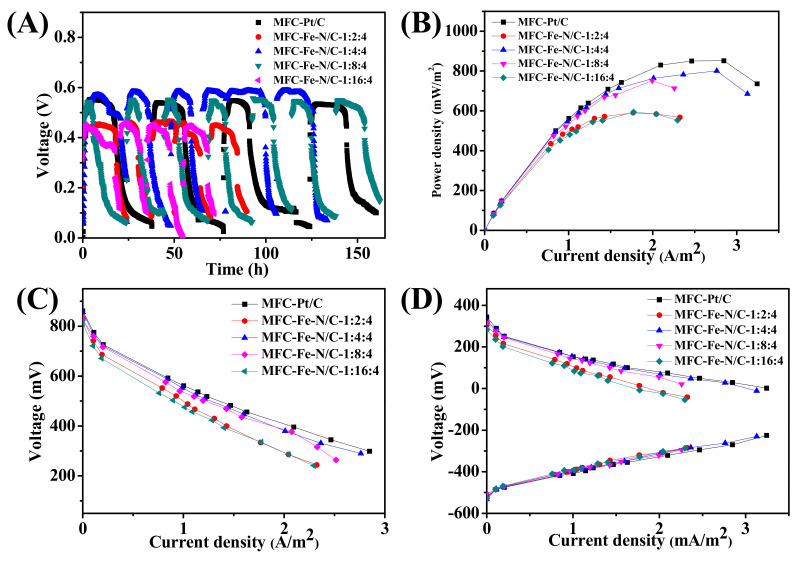
Electricity production performance of the MFCs using the catalysts with different N-doping contents. (**A**) The voltage output variation trend of the MFCs; (**B**,**C**) The power density and polarization curves of the MFCs, respectively; (**D**) The anode and cathode polarization curves.

**Figure 6 nanomaterials-11-00377-f006:**
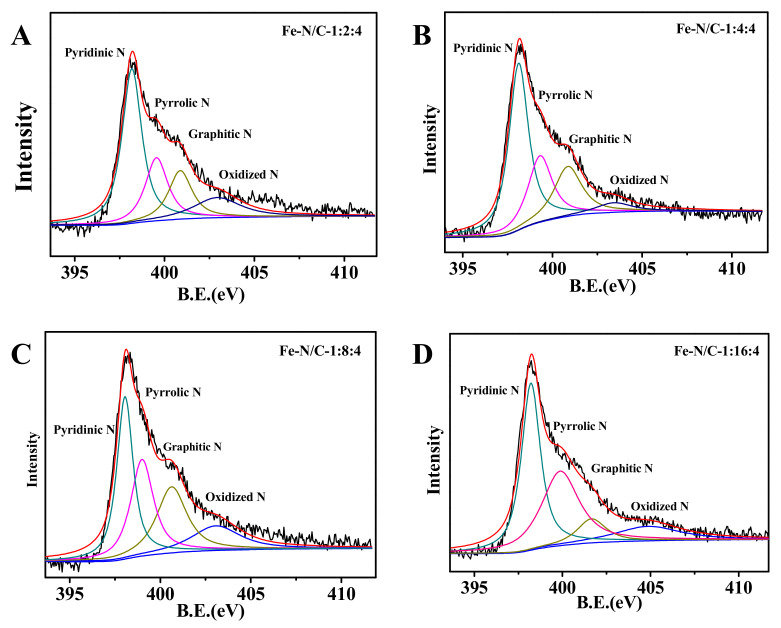
Deconvolution of N1s XPS spectra of: (**A**) Fe-N/C-1:2:4; (**B**) Fe-N/C-1:4:4; (**C**) Fe-N/C-1:8:4; and (**D**) Fe-N/C-1:16:4.

**Figure 7 nanomaterials-11-00377-f007:**
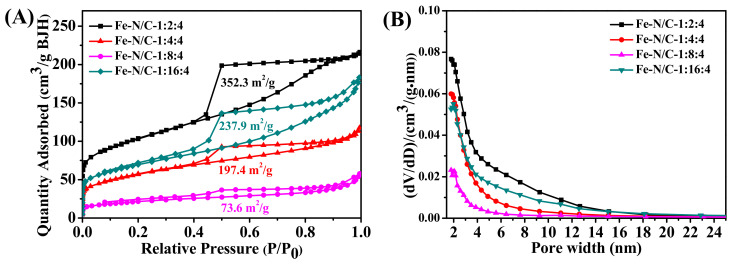
The N_2_ adsorption–desorption isotherms (**A**) and the corresponding pore size distributions (**B**) of Fe-N/C-1:2:4, Fe-N/C-1:4:4, Fe-N/C-1:8:4, and Fe-N/C-1:16:4.

**Figure 8 nanomaterials-11-00377-f008:**
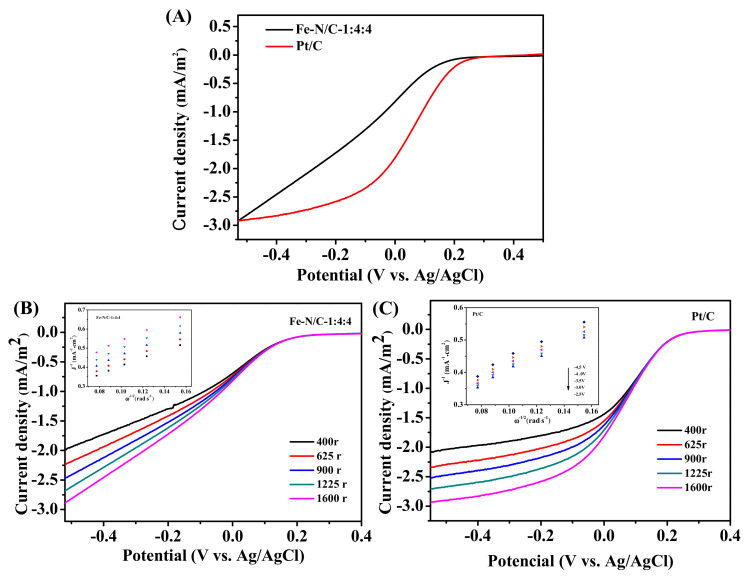
(**A**) The LSV curves of the Fe-N/GNR@CNT-1:4:4 and Pt/C; (**B**,**C**) The LSV curves of Fe-N/C-1:4:4 and Pt/C at different rotating speeds. Insets: K–L plots.

**Table 1 nanomaterials-11-00377-t001:** The content of surface elements and different N species of Fe-N/C-1:2:4, Fe-N/C-1:4:4, Fe-N/C-1:8:4, and Fe-N/C-1:16:4 materials tested by XPS.

	Fe-N/C-1:2:4(at. %)	Fe-N/C-1:4:4(at. %)	Fe-N/C-1:8:4(at. %)	Fe-N/C-1:16:4(at. %)
**C**	85.49	83.94	85.32	85.14
**N**	7.57	8.44	7.95	8.42
**Fe**	1.20	1.64	1.50	1.50
**O**	5.59	5.68	5.03	4.77
**S**	0.15	0.30	0.20	0.16
**Pyridinic-N**	46.67	49.54	32.82	41.95
**Pyrrolic-N**	21.57	24.35	27.54	35.57
**Graphitic-N**	17.29	20.91	24.71	9.14
**Oxidized-N**	14.47	5.20	14.93	12.34

**Table 2 nanomaterials-11-00377-t002:** The summary of the BET specific area of Fe-N/C-1:2:4, Fe-N/C-1:4:4, Fe-N/C-1:8:4, and Fe-N/C-1:16:4.

	Micropore Area (m^2^·g^−1^)	External Surface Area (m^2^·g^−1^)	BET Surface Area (m^2^·g^−1^)
**Fe-N/C-1:2:4**	79.4	272.9	352.3
**Fe-N/C-1:4:4**	22.5	174.9	197.4
**Fe-N/C-1:8:4**	9.7	63.9	73.6
**Fe-N/C-1:16:4**	43.7	194.2	237.9

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
