# Peer review of "Preparation and Application of Fe-N Co-Doped GNR@CNT Cathode Oxygen Reduction Reaction Catalyst in Microbial Fuel Cells"

_nanomaterials, 2021, doi:10.3390/nano11020377_

Round 1

Reviewer 1 Report

I think that this work is an interesting work, and it contributes to better understanding to perform and preparation of Fe/N-doped graphene/CNTs as an air-cathode catalyst in microbial fuel cells. The work seems to have been well planned and performed and, in that sense, the manuscript is suitable for publication in Nanomaterials after some changes for improving understanding and comprehension:

In this work show the performance of microbial fuel cells (MFCs) using Fe/n doped graphene CNTS (1:4:4) as cathode in MFC, “The results showed that MFC with this cathode obtain a maximum power density of 800 mW/m2 and current density around 3 A/m2. The authors must include more reference in this part of results. I suggest to author:

Discuss the advantages of the optimization chamber microbial fuel cells with other works as Wang et al., 2018 (1080 mW/m2), Penteado et al., 2017(420 mW/m2) and Asensio et al., 2018(268 mW/m2). Explain why in this case greater power (W/m2) is achieved than in previous works, explain the influence of cathod materials, and I suggest that the authors shows the different of prices. I recommend the authors to add more detailed discussion about methods section. Also more explanation about the difference with the our previous work (Wang et al., 2018).

Wang, D., Ma, Z., Xie, Y., (...), Zhao, N., Song, H. (2018). Fe/N-doped graphene with rod-like CNTs as an air-cathode catalyst in microbial fuel cells. RSC Advances 8(3), pp. 1203-1209

Penteado, E.D. Fernandez-Marchante, C.M., Zaiat, M., Cañizares, P., Gonzalez E.R., Rodrigo M.A. Influence of carbon electrode material on energy recovery from winery wastewater using a dual-chamber microbial fuel cell. Environmental Technology 38 (2017) 1333-1341.

Asensio, Y., Fernandez-Marchante, C.M., Lobato, J., Cañizares, P., Rodrigo, M.A. Influence of the ion-exchange membrane on the performance of double-compartment microbial fuel cells. Journal of Electroanalytical Chemistry 808 (2018) 427-432.

Author Response

 Dear reviewer,

A point-by-point response to the reviewer’s comments was uploading as a word file. Please check it. Wish good!

Reviewer 2 Report

Authors in this study prepared a Fe-N doped GNR@CNT catalyst as a possible ORR catalyst in microbial fuel cells. The introduction is nicely drafted and experiments are carefully designed. Overall the manuscript is well-drafted and the results obtained are also of significance. I would suggest the paper to be accepted after minor comments. 

  1. Authors can show the TEM images of the Fe-N catalysts and the distribution of Fe nanoparticles on the GNR@CNT surface and related discussion needs to be added.
  2. It is suggested to show the LSV at different rpms and K-L plots to calculate the number of electrons. 
  3. Reviewer strongly suggests the authors avoid using Pt as counter electrons and replace it with the carbon rod electrode to avoid the interference of any leached Pt.
  4. Authors can compare the power density of Fe catalyst with the other non-precious catalysts reported in the literature. The reviewer asked the authors to refer to recent literature for a comparative table (10.1016/j.ijhydene.2020.07.252 and 10.3390/catal10050475)

Author Response

Dear reviewer,

A point-by-point response to the reviewer’s comments was uploaded as a word file. Please check it. Best wishes!

Reviewer 3 Report

In this article, the authors have synthesized GNR@CNT materials with abundant edges by chemical oxidation which showed potential activity for ORR. The author claimed that observed reactivity can be attributed to the synergistic effect of Fe3C nanoparticles, pyridinic- and graphitic-N groups, and oxygen-containing species. Overall this MS is well written and suitable for publication.

I have few minor changes and I would like to see this in the revised MS.

  1. Please include a table of comparison highlighting the Iron-carbon-based ORR catalyst with other best electrocatalysts. Include some comparison based on this in the introduction.
  2. In this statement "Until now, a range of carbon materials have been explored containing hierarchically porous carbon,[28] carbon nanosheets,[29] carbon nanotubes (CNTs),[30] and graphene,[31], please add recent research on CTF based electrocatalyst. As an example, you can find an interesting article published recently in ACS Appl. Mater. Interfaces 2020, 12, 40, 44689–44699 which also explain N-py and N-quart on ORR.
  3. Please move XPS-N spectra, BET, and PSD (Figure S6A1-A4, Figure S7A).[Figure S7B) to the main text and rearrange the ESI.

Author Response

Dear reviewer,

A point-by-point response to the reviewer’s comments was uploaded as a word file. Please check it. Best wishes!

Prof. Song
